# A Robust Planar Marker-Based Visual SLAM

**DOI:** 10.3390/s23020917

**Published:** 2023-01-13

**Authors:** Zhoubo Wang, Zhenhai Zhang, Wei Zhu, Xuehai Hu, Hongbin Deng, Guang He, Xiao Kang

**Affiliations:** 1School of Mechatronical Engineering, Beijing Institute of Technology, Beijing 100081, China; 2UV Center, China North Vehicle Research Institute, Beijing 100072, China

**Keywords:** visual SLAM, planar markers, pose ambiguity

## Abstract

Many visual SLAM systems are generally solved using natural landmarks or optical flow. However, due to textureless areas, illumination change or motion blur, they often acquire poor camera poses or even fail to track. Additionally, they cannot obtain camera poses with a metric scale in the monocular case. In some cases (such as when calibrating the extrinsic parameters of camera-IMU), we prefer to sacrifice the flexibility of such methods to improve accuracy and robustness by using artificial landmarks. This paper proposes enhancements to the traditional SPM-SLAM, which is a system that aims to build a map of markers and simultaneously localize the camera pose. By placing the markers in the surrounding environment, the system can run stably and obtain accurate camera poses. To improve robustness and accuracy in the case of rotational movements, we improve the initialization, keyframes insertion and relocalization. Additionally, we propose a novel method to estimate marker poses from a set of images to solve the problem of planar-marker pose ambiguity. Compared with the state-of-art, the experiments show that our system achieves better accuracy in most public sequences and is more robust than SPM-SLAM under rotational movements. Finally, the open-source code is publicly available and can be found at GitHub.

## 1. Introduction

Multi-sensor fusion has attracted the interest of SLAM researchers in the field of autonomous robots. VINS [1], ORB-SLAM3 [2], and Smart Markers [3] are excellent visual–inertial systems. However, the precise extrinsic parameters of camera-IMU are a prerequisite for these systems. Although some systems (such as VINS and VI-ORB-SLAM [4]) provide online calibration methods, they often fail to calibrate the extrinsic parameters of a camera-IMU mounted on vehicles. This is because the vehicle generally drives on a flat surface and does not generate enough excitation. In this case, accurate camera poses are beneficial, allowing for us to calibrate the external parameters of camera-IMU using the hand–eye method [5]. Many outstanding visual SLAM systems can be used to obtain camera poses, such as ORB-SLAM2 [6], LSD-SLAM [7], and LDSO [8]. However, these methods, based on natural features or optical flow, could fail under illumination conditions, or instances with high dynamics, vigorous rotation, or a low-texture environment [9]. Additionally, they cannot obtain camera poses with a metric scale in the monocular case. Therefore, we prefer to sacrifice the flexibility of such methods to improve accuracy and robustness using artificial planar landmarks.

The planar marker is an easy-to-obtain artificial mark. We can easily print the ArUco [10,11] or AprilTag [12] markers in our laboratory. Therefore, planar-marker-based SfM or SLAM are very popular [13,14,15,16,17]. Although marker-based SLAM is much simpler than feature-based SLAM, a challenging problem is that the planar marker contains ambiguity. The pose is the relative pose between the camera and the marker. Due to the noise in the four corners of the detection marker, two different solutions will be estimated in practice, and ambiguity only occurs in the rotational components of the pose [18,19,20]. Generally, we use the ’infinitesimal plane-based pose estimation’ (IPPE) method [20] to obtain the two solutions, which include the correct one. Figure 1 shows two solutions $\xi^{\prime }$ (the red bounding box) and $\xi^{\prime \prime }$ (the green bounding box), which are returned by IPPE method for a planar marker with ID 205. The red bounding box is the correct solution. The literature [14,15] has proposed several means to distinguish the correct solution. Nevertheless, these methods are still not robust enough and have difficulties in rotational movements.

To avoid or alleviate the difficulties in rotational movements, we propose enhancements to the traditional SPM-SLAM system [15]. We present an improved version of this existing algorithm. We mainly improve four steps in SPM-SLAM. First, in the one-frame initialization, the correct maker pose is only chosen by IPPE if the marker is within a suitable view of the camera and the *ratio* from Equation (Equation 4) is lower than the threshold (Section 4.1). The two-frame initialization is successful if the distance or angle between two frames is greater than the corresponding threshold (Section 4.1). This makes it easier for the initialization to succeed under rotational movements. Second, the current frame can be inserted into the map as a keyframe if the angle between the current frame and the reference keyframe is greater than the threshold. This ensures that our system inserts enough keyframes under the rotation movements to continue tracking (Section 4.2). Third, we present a new approach to select the best marker pose from a set of images by minimizing the M2M error (Section 4.3). Finally, during the relocalization process, when one marker is detected, its correct pose is chosen using the same method as the one-frame initialization approach. When multiple markers are detected, their correct poses are chosen by minimizing the M2M error (Section 4.4). We tested our system on the public dataset and our dataset. Compared with SPM-SLAM [15], UcoSLAM [16], and TagSLAM [17], our system achieves better accuracy in most public sequences, and the speed of our system is much faster than that of UcoSLAM and TagSLAM. The results of our dataset show that our method is more robust and obtains more planar markers in the map than SPM-SLAM under rotational movements. Additionally, the open-source code is available on GitHub.

## 2. Related Research

As our work is dedicated to the planar marker-based SLAM system, a review of the natural feature-based SLAM literature is beyond the scope of this paper. Both Ref. [9] and Ref. [21] contain detailed reviews of natural feature-based SLAM systems. This section will focus on the marker-based SLAM and the problem of marker pose ambiguity.

SPM-SLAM [15] is the first real-time marker-based SLAM system. It is able to deal with ambiguity problems and operate in large indoor environments. The SLAM system sequentially runs the tracking, mapping, and loop-closing processes in a single thread and achieves a good performance in most environments. However, our experiments (Section 5.2) indicated that SPM-SLAM often failed to locate in certain situations where the camera’s motion included rotational movements. UcoSLAM [16] is an extended version of SPM-SLAM and fuses the keypoints with squared fiducial markers. When disabling keypoints, it is similar to SPM-SLAM. However, it needs more computational resources and the addition of keypoints did not significantly improve SPM-SLAM’s tracking accuracy in our experiments.TagSLAM [17] is also a visual SLAM with AprilTag fiducial markers. It considers every frame to be a keyframe and relies on iSAM2 [22], which uses a factor graph to represent the pose optimization problems. However, TagSLAM cannot run in real-time unless a trusted map of tag poses is already available, because the graph grows over time and the CPU load increases. Our experiments indicated that TagSLAM is unstable. MarkerMapper [14] is an offline method that obtains a map of fiducial markers and then obtains camera poses. It has several limitations. The most serious one is that the proposed method cannot run in real-time.

Additionally, the *ratio* test [20] is a general method to resolve the planar-marker’s pose ambiguity. Marker-based SfM [14], SPM-SLAM [15], and UcoSLAM [16] also apply this method. The *ratio* test means that if the *ratio* from Equation (Equation 4) is below a certain threshold, the IPPE solution with lower reprojection errors is the correct one. However, this was proved to not always be correct. Unlike the *ratio* test, which used one frame to solve marker pose ambiguity, S. -F. Ch’ng [23] exploited multi-view constraints for disambiguation. This method can efficiently choose the correct pose from two ambiguous poses by formulating a clique-constrained rotation averaging problem and a maximum weighted clique problem. However, the method consumes a high amount of computing resources and is not suitable for SLAM running in real-time.

## 3. System Overview

The enhanced marker-based SLAM is a monocular SLAM system. It is an accurate and real-time system used to estimate camera poses and create a map of planar markers. To describe the improvements that we have made, Section 3.1 explains some notations that were employed in this work. For completeness, Section 3.2 provides a brief description of the operational information of SPM-SLAM. Section 3.3 describes some improvements that we made.

### 3.1. Concepts and Notations

Figure 2 shows the relationship between the three reference systems and defines some terms employed in this work. Let us denote $f^{t}$ as a frame captured by a camera at the time *t*. The equations of pose transform between the three reference systems are represented by
(1)$$\left\{\begin{array}{c} \begin{array}{cc} T_{cw}^{t} & =T_{cm}^{t}\cdot {\left(T_{wm}\right)}^{-1}\\ T_{wm} & ={\left(T_{cw}^{t}\right)}^{-1}\cdot T_{cm}^{t}\\ T_{cm}^{t} & =T_{cw}^{t}\cdot T_{wm} \end{array} \end{array}\right.,$$
where $T_{cw}^{t}\in \mathrm{SE}\left(3\right)$ is the pose that transforms points from the world reference system (*wrs*) to the camera reference system (*crs*) at the time *t*. SE(3) [24] is the group of rigid transformations in 3D space and a 4×4 matrix. $T_{wm}\in \mathrm{SE}\left(3\right)$ is the pose that transforms points from the marker reference system (*mrs*) to the world reference system. $T_{cm}^{t}\in \mathrm{SE}\left(3\right)$ is the pose that transforms points from the marker reference system to the camera reference system at the time *t*.

We also denote the three-dimensional point $p_{i}\in R^{3}\left(i=1,2,3,4\right)$ as the four corners of marker *m* in the marker reference system, and $u_{i}^{t}\in R^{2}$ as the pixel locations observed in a frame $f^{t}$. Let us denote ${\tilde{u}}_{i}^{t}$ as the projection that can be represented by
(2)$${\tilde{u}}_{i}^{t}=\Psi \left(K,D,T_{cm}^{t},p_{i}\right),$$
where ***K*** and ***D*** are the camera-intrinsic parameters and distortion coefficients. Naturally, given the transform matrix $T_{cm}^{t}$, the reprojection error of a marker’s corners in a frame $f^{t}$ is calculated as:(3)$$e_{m}^{t}\left(T_{cm}^{t}\right)=\sum_{i=1}^{4}{∥\Psi \left(K,D,T_{cm}^{t},p_{i}\right)-u_{i}^{t}∥}^{2}.$$

As mentioned earlier in Section 1, the pose of the marker *m* observed in the frame $f^{t}$ has two solutions because of marker pose ambiguity. We denote ${\dot{T}}_{cm}^{t}$ and ${\ddot{T}}_{cm}^{t}$ as the two solutions returned by the IPPE method. The reprojection error *ratio* caused by the poses ${\dot{T}}_{cm}^{t}$ and ${\ddot{T}}_{cm}^{t}$ is represented by
(4)$$ratio=\frac{\min\left(e_{m}^{t}\left({\dot{T}}_{cm}^{t}\right),e_{m}^{t}\left({\ddot{T}}_{cm}^{t}\right)\right)}{\max\left(e_{m}^{t}\left({\dot{T}}_{cm}^{t}\right),e_{m}^{t}\left({\ddot{T}}_{cm}^{t}\right)\right)}.$$

We denote the angle $\theta_{m}^{t}$ as the view angle by which the marker *m* is observed in the frame $f^{t}$. Considering the translation components $t_{cm}^{t}$ of the pose $T_{cm}^{t}$, we calculate the angle $\theta_{m}^{t}$ by
(5)$$\theta_{m}^{t}=\arccos\frac{z_{m}^{t}}{\sqrt{{\left(x_{m}^{t}\right)}^{2}+{\left(y_{m}^{t}\right)}^{2}+{\left(z_{m}^{t}\right)}^{2}}}.$$
where $t_{cm}^{t}=\left(x_{m}^{t},y_{m}^{t},z_{m}^{t}\right)$. Since marker pose ambiguity only exists in the rotational components of the pose, the translation components of two solutions (${\dot{T}}_{cm}^{t}$ and ${\ddot{T}}_{cm}^{t}$) are available in (Equation 5). Then, the angle $\theta_{12}$ between the two frames is represented as:(6)$$\theta_{12}=\frac{\sum_{i=1}^{n}\left|\theta_{i}^{t1}-\theta_{i}^{t2}\right|}{n}.$$
where *i* refers to *n* markers that can be observed in both frames.

### 3.2. SPM-SLAM

In this section, we review some of the SPM-SLAM algorithms, which are essential to the proposed improvements in this paper. As we can observe from Figure 3, SPM-SLAM runs in a single thread. Firstly, it localizes the camera with every frame by finding ArUco tag matches and minimizing the error that derives from reprojecting these tags to the previous frame.Then, it localizes the new tags from a set of ambiguous observations. After that, it uses local bundle adjustment to optimize a set of keyframes and ArUco tags. Finally, once a revisited tag is detected, the system corrects the accumulated drift using pose-graph optimization with the g2o framework [25]. After pose-graph optimization, it performs global optimization and optimizes the poses of tags and keyframes.

### 3.3. Improvements in Our System

This section mainly describes the operational process of the system and the improvements we made. Figure 3 describes the system’s operational information. The green bounding boxes are the improvement methods that we proposed. ArUco or AprilTag markers can be detected in our system by the ArUco library [10,11] or AprilTag 3 library [26].

At first, the initialization process establishes the world reference system for the map and adds marker(s) and keyframe(s) to the map. Our improved initialization method (Section 4.1) keeps the process more robust in terms of its rotational movement. After successful initialization, the system enters tracking mode. The tracking process aims to estimate the current frame pose $T_{cw}^{t}$ by minimizing the reprojection error of the visible markers’ corners for each new captured frame $f^{t}$ [15]. When tracking failure occurs, relocalization is started. Subsequently, the system performs the keyframe decision process. The method of inserting the keyframe and makers to the map is introduced in Section 4.2. The pose(s) $T_{wm}$ of new marker(s) will be estimated later by a new method. This is one of the contributions of this paper. The method estimates the marker pose from multi-view frames by minimizing the average rotation between markers (Section 4.3). After the insertion of keyframes and markers, a keyframe culling process is run to delete unnecessary keyframes in the map, which is reduces the processing time of local optimization and global optimization. Local optimization aims to simultaneously optimize the keyframe and marker poses by minimizing the markers’ reprojection errors.

The loop closing process is run after the tracking process. It effectively eliminates the errors accumulated along the path [27]. The final step before a new frame is captured is selecting a reference keyframe $f^{w}$ from the map for the current frame $f^{t}$. The process of tracking, loop closure, and local optimization for the next frame $f^{t+1}$ will use the reference keyframe. This is the one that is nearest to the current frame $f^{t}$. Finally, the global optimization process is run at the end of the system. All the keyframes and markers of the map are employed in this.

## 4. Enhanced Methods

### 4.1. Initialization

The initialization process aims to establish a world reference system for the map, and add keyframes and markers with correct poses to the map, which is vital to an SLAM system. As in SPM-SLAM, we adopt one-frame and two-frame initialization. However, we set different conditions to decide whether the process succeeds.

One-frame initialization succeeds if the *ratio* of at least one marker is lower than the threshold $\tau_{e}$ in SPM-SLAM [15]. However, when the marker is on the edge of an image, it is sometimes misjudged. In our system, the following two conditions must be met for one-frame initialization to succeed.
(7)$$ratio<\tau_{e},\;and\;\theta_{m}^{t}<\tau_{\theta },$$
where $\theta_{m}^{t}$ is calculated by (Equation 5). The condition ($\theta_{m}^{t}<\tau_{\theta }$) means that a marker *m* must be observed within a proper scope of view. The value $\tau_{\theta }$ is related to the camera’s angle of view (FOV). This is set to 0.4–0.6 times the camera FOV used in our experiment.

For two-frame initialization, our method is the same as that of SPM-SLAM, which estimates the pose of frames and markers that are ambiguously observed in the two frames [15]. Two-frame initialization is successful if one of the following two conditions is met.
(8)$$d\left(f^{0},f^{1}\right)>\tau_{b},\;or\;\theta_{12}>\Delta \theta ,$$
where $d(f^{0},f^{1})$ is the distance between $f^{0}$ and $f^{1}$. The angle $\theta_{12}$ between $f^{0}$ and $f^{1}$ is calculated by (Equation 6). In contrast to SPM-SLAM’s method, the condition $\theta_{12}>\Delta \theta$ helps our system to successfully initialize under rotational movements.

### 4.2. New Keyframes and Markers Insertion

When a frame pose $T_{cw}^{t}$ is successfully estimated in the tracking process, we need to decide whether the frame can be judged as a keyframe and inserted into the map. The insertion of keyframes has a heavy influence on local optimization and tracking. Too many keyframes may cause local optimization to consume a high amount of computational resources, while fewer keyframes may lead to a poor tracking performance, or even tracking failure. Therefore, keyframes can be inserted into the map when one of the following conditions are satisfied:If at least one new marker is detected in a frame, the frame is spawned as a new keyframe and added to the map with the new marker(s). Notably, the pose(s) of the new marker(s) are set to zero. Later, these are estimated from multiple views (Section 4.3).If the distance between the current frame $f^{t}$ and the reference keyframe $f^{w}$ is larger than the threshold $\tau_{b}$, the frame is added to the map.If the angle between the current frame $f^{t}$ and the reference keyframe $f^{w}$ is larger than the threshold Δθ, the frame is added to the map.

Unlike SPM-SLAM, which only uses conditions 1 and 2, the use of condition 3 ensures that our system instantly inserts new keyframes during rotational movements. This improves the robustness of the system, allowing for it to adapt to more environments.

### 4.3. Marker Pose Estimation

Once a new marker *m* is added to the map, the marker poses $T_{wm}$ should rapidly be obtained. We propose a novel method to estimate the marker pose from a set of keyframes that observe that marker *m*. Let us denote F as the set of keyframes. The poses of F with keyframes are represented by $\left(T_{cw}^{1},T_{cw}^{2},\cdots ,T_{cw}^{n}\right)$, which are known after the tracking process. Considering the pose ambiguity of marker *m* in the set of keyframes F, we can obtain possible poses for the marker *m* according to (Equation 1), which is denoted by:(9)$$\Phi \left(m\right)=\left\{{\dot{T}}_{wm}^{1},{\ddot{T}}_{wm}^{1},\cdots ,\;{\dot{T}}_{wm}^{n},{\ddot{T}}_{wm}^{n}\right\}$$

Our goal is to choose the best pose $T_{wm}$ of the marker *m* from Φ(m). Given a set of F, the relative rotations from marker *i* to marker *j* (M2M) have four solutions, which are represented by
(10)$$\left\{\begin{array}{c} \begin{array}{cc} R_{i,j}^{t,00} & ={\left({\dot{R}}_{j}^{t}\right)}^{T}\cdot {\dot{R}}_{i}^{t}\\ R_{i,j}^{t,01} & ={\left({\ddot{R}}_{j}^{t}\right)}^{T}\cdot {\dot{R}}_{i}^{t}\\ R_{i,j}^{t,10} & ={\left({\dot{R}}_{j}^{t}\right)}^{T}\cdot {\ddot{R}}_{i}^{t}\\ R_{i,j}^{t,11} & ={\left({\ddot{R}}_{j}^{t}\right)}^{T}\cdot {\ddot{R}}_{i}^{t} \end{array} \end{array}\right.,$$
where $\left\{{\dot{R}}_{i}^{t},{\ddot{R}}_{i}^{t}\right\}$ are the rotational components of the two ambiguous poses of marker *i* in the frame $f^{t}$. They have a 3×3 matrix. Let us denote ${\left\{M_{i}\right\}}_{i=1}^{p}$ as the observed markers with the correct poses from the set of keyframes F. Remember that these poses are known, and that we denote $\left\{R_{w1},\cdots ,R_{wp}\right\}$ as the rotational components.

Our method is to select the best pose $T_{wm}$ from Φ(m) by minimizing the M2M error between marker *m* and markers ${\left\{M_{i}\right\}}_{i=1}^{p}$ in the set of keyframes F. The M2M error is represented by
(11)$$e\left(T_{wm}\right)=\sum_{t=1}^{n}\sum_{i=1}^{p}\min\left(\begin{array}{cc} \; & ∥{\left(R_{wi}\right)}^{T}\cdot R_{wm}-R_{m,i}^{t,00}{∥}_{F}^{2}\\ \; & ∥{\left(R_{wi}\right)}^{T}\cdot R_{wm}-R_{m,i}^{t,01}{∥}_{F}^{2}\\ \; & ∥{\left(R_{wi}\right)}^{T}\cdot R_{wm}-R_{m,i}^{t,10}{∥}_{F}^{2}\\ \; & ∥{\left(R_{wi}\right)}^{T}\cdot R_{wm}-R_{m,i}^{t,11}{∥}_{F}^{2} \end{array}\right)$$
where $R_{wm}$ is the rotational components of $T_{wm}$, and ${∥\cdot ∥}_{F}^{2}$ denotes the Frobenius norm. The best solution is the one that minimizes the M2M error e($T_{wm}$) in the set of keyframes F. Note that the translation component is already included in $T_{wm}$. Therefore, we can find a correct pose for marker *m* by (Equation 11). Furthermore, the angle or distance between the keyframes in F must be larger than threshold Δθ or threshold $\tau_{b}$, respectively. This could make (Equation 11) more effective. Note that the parameters Δθ and $\tau_{b}$ are the same as those mentioned in Section 4.2.

### 4.4. Relocalization

The relocalization is run when the system fails to track. Let us denote M as a marker with the correct pose $T_{wm}$ detected in current frame $f^{t}$, and *n* is the number of markers in M.

If n<1, the process will fail.

If n=1, two methods can be used to obtain a solution. One is that, if the marker satisfies condition (Equation 7), we can obtain the marker’s unambiguous pose $T_{cm}^{t}$. Then, we can successfully compute the current frame pose $T_{cw}^{t}$ using (Equation 1). The other method is that, if the marker is not satisfied with the condition (Equation 7), we can obtain the marker’s ambiguous poses ${\dot{T}}_{cm}^{t}$ and ${\ddot{T}}_{cm}^{t}$ using (Equation 1). Naturally, the two poses (${\dot{T}}_{cw}^{t}$ and ${\ddot{T}}_{cw}^{t}$) of the current frame are computed. Then, the two distances between the current frame and the reference frame are calculated using the two poses. Finally, the pose with the smaller distance is the correct one.

If n>1, let us denote
(12)$$\Omega \left(T_{cn}^{t}\right)=\left\{{\dot{T}}_{c1}^{t},{\ddot{T}}_{c1}^{t},\cdots ,\;{\dot{T}}_{cn}^{t},{\ddot{T}}_{cn}^{t}\right\},$$
as the ambiguous poses of markers M in current frame $f^{t}$, and denote
(13)$$\Upsilon \left(T_{wn}\right)=\left\{T_{w1},\cdots ,T_{wn}\right\},$$
as the correct poses. The M2M error in M is represented by:(14)$$\underset{i<j<n}{e<i,j>}={∥{\left(R_{ci}\right)}^{T}\cdot R_{cj}-{\left(R_{wi}\right)}^{T}\cdot R_{wj}∥}_{F}^{2},$$
where $R_{ci}$ (3×3 matrix) is the rotational component of $T_{cn}^{t}\in \Omega \left(T_{cn}^{t}\right)$, and $R_{wi}$ is the rotational component of $T_{wn}\in \Upsilon \left(T_{wn}\right)$. Here, note that if the right rotational component is selected, the translation component is also selected. Accordingly, our goal is firstly to select a pair of poses $T_{ci}^{t}$ and $T_{cj}^{t}$ from $\Omega (T_{cn}^{t})$ by minimizing the M2M error (Equation 14), and then calculate the pose with (Equation 1). As the correct poses ($\left\{T_{wi},\;T_{wj}\right\}\in \Upsilon \left(T_{wn}\right)$) are known, we can obtain two solutions ($T_{cw_{i}}^{t}$ and $T_{cw_{j}}^{t}$) for the current frame pose. If the distance between $T_{cw_{i}}^{t}$ and $T_{cw_{j}}^{t}$ is lower than the threshold $\sigma_{e}$ (default is 0.01m in our system), the relocalization process will succeed and the two poses ($T_{cw_{i}}^{t}$ and $T_{cw_{j}}^{t}$) are available. Otherwise, relocalization will fail.

## 5. Experiments and Results

This section evaluates our method’s performance in the publicly available SPM dataset [15] and our dataset. The SPM dataset has eight sequences and provides ground truth trajectories for each sequence. Our dataset contains six indoor sequences and two outdoor sequences, which include more rotational movements. We aim to evaluate the tracking accuracy and robustness of our methods. Since the SPM dataset offers the ground truth, it was used to evaluate the tracking accuracy. We assessed the robustness of our system on our dataset, which includes many rotational movements. All tests were run on an Intel Core i5-11600K 3.9GHz desktop computer.

In addition, our method can be compared with other marker-based SLAM systems, which include SPM-SLAM, TagSLAM, and UcoSLAM, as SPM-SLAM, UcoSLAM, and our system have the same parameters. In order to be comparable, these same parameters must be equal. The parameters $\tau_{b}$ and $\tau_{e}$ (Section 4) employed in all three systems were set at 7 mm and 0.333 in the two datasets. The other parameters Δθ and $\sigma_{e}$, which were only employed in our system, were set at 5° and 0.01 m, which proved to be good values in most of our experiments.

### 5.1. SPM Dataset

The SPM dataset contains eight video sequences, with fiducial markers recorded in a laboratory, and provides a ground truth. The camera works at 60 Hz with a resolution after rectification of 1920 × 1080 pixels. The sequences 01, 02, 03, 07, and 08 were recorded by a camera pointed toward the walls, and others were recorded by a camera pointed towards the ceiling. We evaluated the accuracy of the trajectory using the Absolute Trajectory Error (ATE) measure, which is the translational RMSE. In addition, the system’s mean speed was also considered as an evaluation indicator.

SPM-SLAM, TagSLAM, and UcoSLAM were selected for comparison in the experiments. Table 1 shows the ATEs and Trck of the methods tested on the eight sequences. Figure 4 shows the trajectories for ground truth (dash gray), SPM-SLAM (blue), TagSLAM (green), UcoSLAM (red), and ours (purple). As can be observed, TagSLAM’s accuracy was the worst, especially in sequence 01, where its ATE was 0.43 m. Compared with SPM-SLAM and UcoSLAM, our method’s accuracy was slightly better in most sequences. Figure 5 shows the mean speed of the four systems. The graph shows that TagSLAM and UcoSLAM are much slower than other methods. This is because TagSLAM considers every frame to be a keyframe and UcoSLAM combines keypoints and markers, forming the CPU load. In contrast, our method and SPM-SLAM can run at about 145 frames per second (fps) on the SPM dataset.

### 5.2. Our Dataset

The test on the public dataset showed that our method significantly outperforms TagSLAM and UcoSLAM in accuracy and speed. However, the accuracy of our method is slightly better than SPM-SLAM in most sequences, and the speed is almost the same as SPM-SLAM. Therefore, to further evaluate that our method is more robust than SPM-SLAM in terms of rotational movements, it was tested and compared with SPM-SLAM on our dataset.

For this test, we recorded eight video sequences. Sequences 01–06 were recorded in a laboratory, and 22 markers were placed on the wall in an approximated dimension of 1.5 × 1.5 square meters. Sequences 07 and 08 were recorded outdoors, and 48 markers were placed on the ground in an approximated dimension of 20 × 15 and 30 × 20 square meters. Figure 6 shows the markers on the ground for sequences 07 and 08. It is worth noting that outdoor sequences 07 and 08 were recorded with a camera mounted on a vehicle. The camera captured images at a 1920 × 1080 pixel resolution and a frame rate of 30Hz. For sequences 01 and 02, the camera was pointed towards the wall and was spinning. Furthermore, sequence 01 was rotated about 270 degrees, and sequence 02 started and ended at the same position to facilitate loop-closure detection. Different from sequences 01–02, sequences 03–06 have more positional movements and only have rotational movements at the corners. For outdoor sequences, sequence 08 showed more rotational movement than sequence 07. Sequences 02–08 can facilitate the detection of loop closure. Although we could not obtain the ground truth for the camera poses, the tracking success rate (Trck) and the number of markers reconstructed in the mapping were used to evaluate the system’s robustness. The Trck is the ratio of the number of tracked frames to the total number of frames in a sequence.

Table 2 shows the number reconstructed markers and the tracking rate. As can be observed, the SPM-SLAM Trcks were lower than 55% on sequences 01–03 and 05–06, while the proposed method achieved over 99% on all sequences. Additionally, SPM-SLAM only reconstructed all markers in sequence 07. In contrast, our method reconstructed all markers in all sequences. Figure 7 shows the reconstruction results of both methods for sequences 01 and 02. This clearly shows that more markers were reconstructed by our method than SPM-SLAM. Figure 8a–h show the reconstructed trajectory of both methods for sequences 03–06. We can observe that our method outperforms SPM-SLAM, especially in sequences 03 and 06, where SPM-SLAM almost completely failed to reconstruct the trajectory. Figure 8i–l show the reconstructed trajectory of both methods for sequences 07 and 08 (outdoors). As can be observed, our method was slightly better than SPM-SLAM on sequence 07 and much better than SPM-SLAM on sequence 08. This is not unexpected, since sequence 08 has more rotational movement than sequence 07. In summary, the experiments show that the robustness of our system is better than that of the SPM-SLAM under rotational movements.

## 6. Conclusions

This paper presents an improved version of the existing SPM-SLAM. To improve the robustness and accuracy, we improved the initialization, keyframe insertion and relocalization algorithms based on the SPM-SLAM dataset and proposed a method to solve the problem of planar-marker pose ambiguity. Our system was compared with the state-of-the-art, such as SPM-SLAM, TagSLAM, and UcoSLAM. The results on public datasets show that our system achieves better accuracy in most sequences, and the speed of our system is much faster than that of UcoSLAM and TagSLAM. The results of our datasets show that our method is more robust than SPM-SLAM under rotational movements and reconstructs more planar markers in the map. Additionally, our system runs at approximately 145 Hz using a single thread, which benefits UAVs and mobile robots, as they require fewer computational resources. It is worth mentioning that the open-source code is available on GitHub (https://github.com/BIT-wangzb/Marker-based-SLAM) (accessed on 10 December 2022).

## Figures and Tables

**Figure 1 sensors-23-00917-f001:**
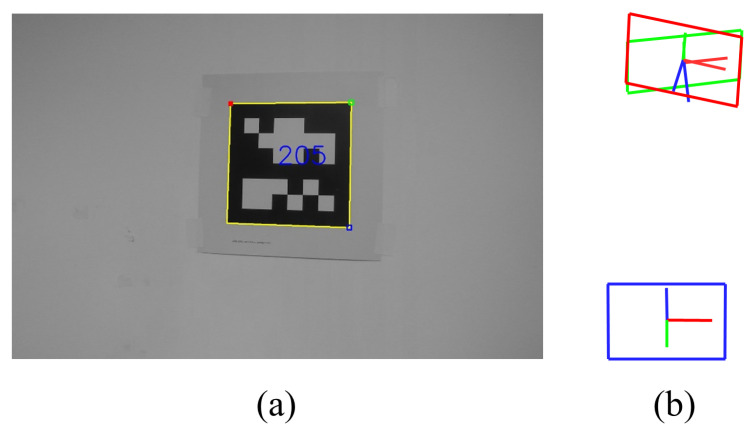
(**a**): A detected marker with ID 205 from an image. (**b**): The two pose solutions, $\xi^{\prime }$ (red bounding box) and $\xi^{\prime \prime }$ (green bounding box), returned by IPPE. The blue bounding box is the camera pose.

**Figure 2 sensors-23-00917-f002:**
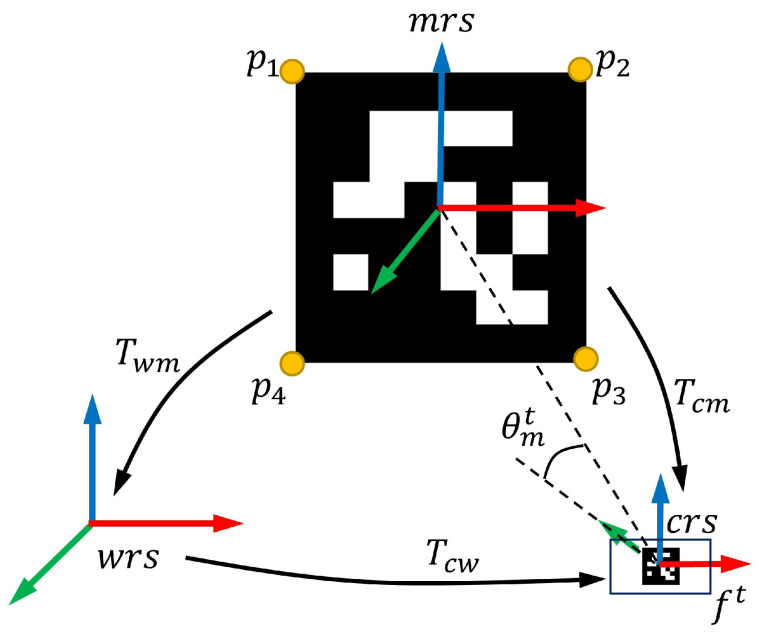
The relationship between three coordinate systems and terms.

**Figure 3 sensors-23-00917-f003:**
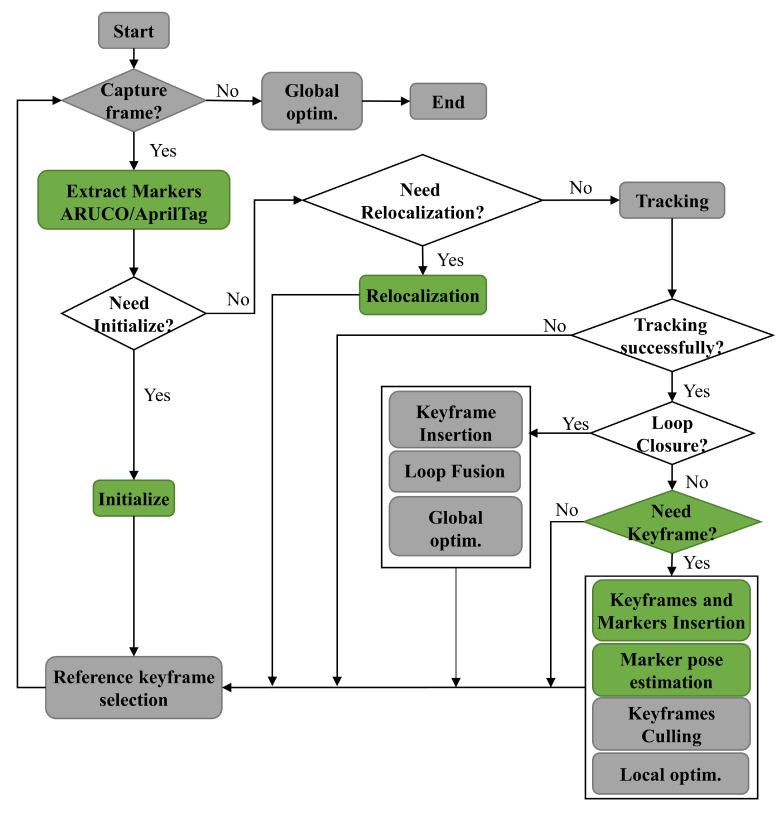
Pipeline for SPM-SLAM and our system. The green bounding boxes are the methods that we improved.

**Figure 4 sensors-23-00917-f004:**
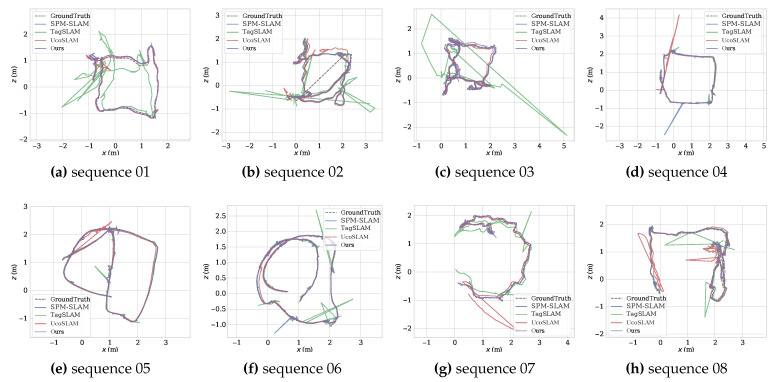
SLAM trajectories in SPM dataset. The dashed gray line is the ground truth. The blue line is SPM-SLAM. The green line is TagSLAM. The red line is UcoSLAM. The purple line is ours.

**Figure 5 sensors-23-00917-f005:**
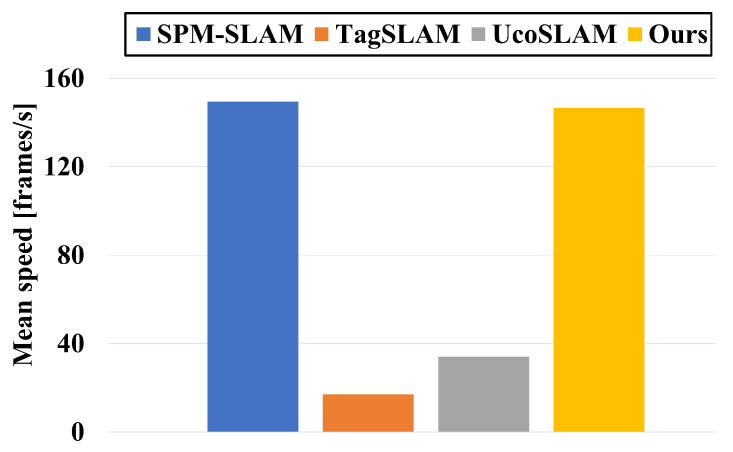
The mean speed of four systems (higher is better).

**Figure 6 sensors-23-00917-f006:**
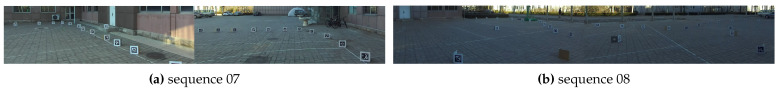
(**a**,**b**) show the markers on the ground for outdoor sequences 07 and 08.

**Figure 7 sensors-23-00917-f007:**
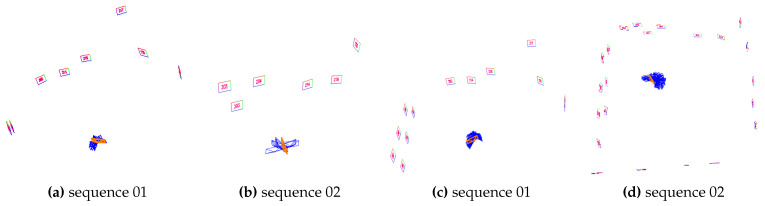
(**a**,**b**) mapping results of SPM-SLAM on sequences 01 and 02. (**c**,**d**) results for our dataset.

**Figure 8 sensors-23-00917-f008:**
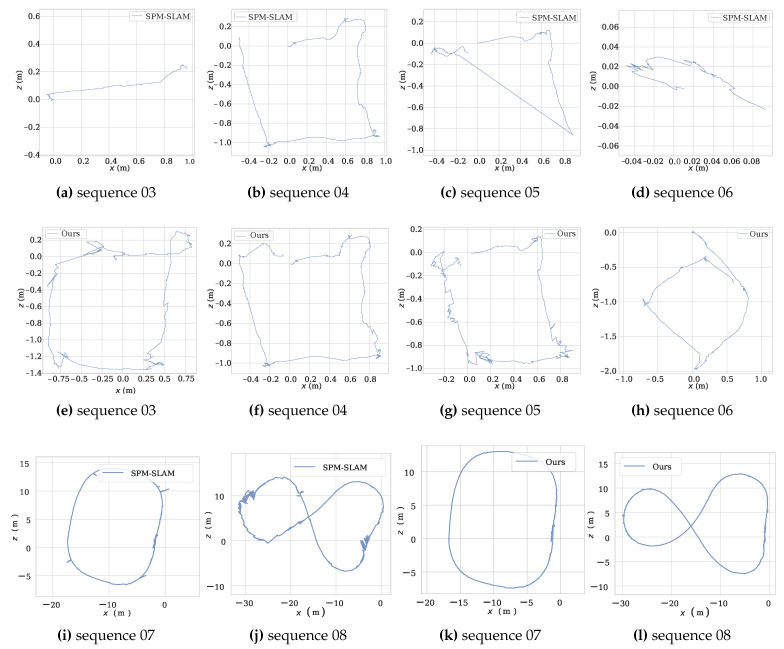
Trajectories on sequences 03-08 in our dataset. (**a**–**d**,**i**,**k**) are generated by SPM-SLAM. (**e**–**h**,**j**,**l**) are generated by ours.

**Table 1 sensors-23-00917-t001:** Absolute trajectory errors on the SPM dataset.

Dataset	Length [m]	SPM-SLAM	TagSLAM	UcoSLAM	Ours
		ATE [m]			
sequence 01	19.5	0.060	0.430	0.084	**0.059**
sequence 02	23.3	0.046	0.054	0.079	**0.045**
sequence 03	23.1	0.055	0.233	0.058	**0.054**
sequence 04	32.8	0.014	0.027	0.037	**0.013**
sequence 05	26.9	0.017	0.023	0.070	**0.016**
sequence 06	33.2	0.017	0.026	0.028	**0.016**
sequence 07	18.9	**0.048**	0.246	0.094	0.050
sequence 08	26.1	0.064	0.066	0.131	**0.063**

**Table 2 sensors-23-00917-t002:** The number of markers reconstructed in the mapping results and tracking success rate of frames.

Dataset	Total Number of Markers	SPM-SLAM	Ours
		Num of Reconstructed Markers/Trck	
sequence 01	12	8/72.6%	**12**/**99.8%**
sequence 02	22	6/13.4%	**22**/**99.9%**
sequence 03	22	8/16.1%	**22**/**99.9%**
sequence 04	22	20/86.2%	**22**/**99.5%**
sequence 05	22	11/54.7%	**22**/**99.9%**
sequence 06	22	2/5.2%	**22**/**99.9%**
sequence 07	48	**48**/99.4%	**48**/**99.9%**
sequence 08	48	47/93.7%	**48**/**99.2%**

## Data Availability

Not applicable.

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
