# Peer review of "A Robust Planar Marker-Based Visual SLAM"

_sensors, 2023, doi:10.3390/s23020917_

Round 1

Reviewer 1 Report

The paper presents an improved version of a marker-based visual SLAM system. 

The approach is well described. The English of the paper is OK - some more proof reading should be done.

Visual SLAM is a very important topic for robotics. Using markers may be a valid approach in some scenarios. 

The reviewer has three major concerns with the paper:

1) The experimental evaluation is lacking for a visual SLAM system: The covered area is tiny! Just in one room in a few meters wide space. What is the application for such a tiny SLAM system? If you really want to use calibration as application you need to show that a calibration algorithm works better by using your approach (but that would be hard to do since most calibration approaches use targets and optimize their position together with the calibration parameters). State of the art visual SLAM systems can handle much bigger areas - e.g. [A] - 9 years ago! 

[A] McDonald, John, et al. "Real-time 6-DOF multi-session visual SLAM over large-scale environments." Robotics and Autonomous Systems 61.10 (2013): 1144-1158.

2) In order to make a really good paper you should compare your approach to state of the art visual SLAM approachtes, too. Like ORB3. 

3) Almost all visual SLAM solutions will have IMU data available. That should help a lot to make your trajectories more smooth/ realistic (looking at 7(g), for example, one can see many jumps that I don't think the camera actually did). Can you elaborate how you could intregrate IMU data into your system?

In the introduction you motivate your work with calibration. But most calibration systems actually do use artificial markers (calibration targets) already. SLAM itself is a good enough motivation - I would put let emphasis on calibration - also because in the end you didn't do any experiments towards calibration.

OK now reading towards the end of your paper I see why you put calibration as an application: your test cases are all tiny.

Why doesn't 4.2 also include tests of TagSLAM and UcoSLAM?

Write: "Generally, we use the 'Infinitesimal plane-based pose estimation' (IPPE) method [16] " - introduce abbreviations first!

For the SPM dataset, we need more info:

 - frame rate

 - length of sequence in terms of time and distance (in m) traveled. Best as a table.

------

Itsn't it strange that the introduction heading starts with the number 0? Shouldn't it be 1?

Some English comments:

Multiple sensors fusion -> Multi-sensor fusion 

line 96: "this section introduces some concepts employed in the system." That is strange!?

line 244: "In the test on the public dataset, we have proved that our method significantly outper- 244 forms TagSLAM UcoSLAM in accuracy and speed." You can't say "prove" in this context! This is a scientific paper - you didn't prove anything! Maybe in another setup your algorithm could fail - so no prove! Use the words "demonstrate" or "showed". Or "validate". 

Reviewer 2 Report

1) The authors have proposed some small improvements over a previous algorithm (some steps of it were modified). I suggest they change the paper accordingly to reflect this plus some small corrections, and I will be satisfied with the paper's quality. It is OK for publication since these corrections/adjustments are performed.

2) Particularly, a discussion on fusion algorithms, e.g. fusing IMU and GPS in a single encapsulation with images (RGBD), should be provided in your manuscript, at least referring to this as further work. It is known that this kind of fusion approach performs better than only using the solely image-based ones. As an example, see the reference below, where a smart marker device and a two-stage algorithm for multimodal sensor fusion and camera pose estimation are proposed (from Sensors): Ortiz-Fernandez, L.E.; Cabrera-Avila, E.V.; Silva, B.M.F.d.; Gonçalves, L.M.G. Smart Artificial Markers for Accurate Visual Mapping and Localization. Sensors 2021, 21, 625. https://doi.org/10.3390/s21020625

3) Optical flow is a method or technique, not a natural landmark: "using natural landmarks such as optical flow or textures".

4) This is not a novel contribution: "This paper presents a monocular visual SLAM using artificial planar markers.".

5) As a scientific writing rule, do not use citations in the abstract: "Our work is an improved version of SPM-SLAM [1].". Instead, put the name of the method or explain it in order for the reader to know what it is about.

6) This has been done in the literature: "we improve the process of the initialization, keyframes insertion, and relocalization.".

7) I'm not sure this is new: "Besides, we propose a novel method to estimate marker poses from a set of images to solve the problem of planar marker pose ambiguity.".

8) Start Introduction with the correct numbering: "0. Introduction" -> "1. Introduction"

9) You put the contribution suddenly, and wrongly (you have not proposed a completely new algorithm). I suggest changing: "This paper proposes a robust planar marker-based SLAM system. It is an improved version of the existing SPM-SLAM system [1]" -> "In order to avoid or alleviate the difficulties in rotational movements pointed above, we propose enhancements in the traditional SPM-SLAM system [1], coming up with an improved version of this existing algorithm.".

10) Change "four algorithms of SPM-SLAM" -> "four steps of SPM-SLAM".

11) Renumbering: "1. Related Research" -> "2. Related Research", and so on.

12) This is not a proposed algorithm, it is an enhancement in an algorithm proposed previously: "The proposed marker-based SLAM". Thus, change the whole paper accordingly to reflect this. Actually, you even agree with this next: "In order to describe the improvements that we have made,...".

13) "3. Proposed Methods" -> "4. Enhanced Methods" (remember, you are proposing enhancements).

14) Correct: "im-prove the robustness" -> "improve the robustness"; "Ad-ditionally, the proposed" -> "Additionally, the proposed".

15) Put the github address also here (or a citation to it in the References Section and cite here): "It is worth mentioning that the open-source code is available on GitHub.".
